# Calibration of Arrhenius Constitutive Equation for B_4_C_p_/6063Al Composites in High Temperatures

**DOI:** 10.3390/ma15186438

**Published:** 2022-09-16

**Authors:** Jian Sun, Yunhui Chen, Fuguang Liu, Erjuan Yang, Sijia Wang, Hanguang Fu, Zhixu Qi, Sheng Huang, Jian Yang, Hui Liu, Xiaole Cheng

**Affiliations:** 1School of Mechanical and Electrical Engineering, Xi’an Polytechnic University, Xi’an 710048, China; 2Xi’an Key Laboratory of Modern Intelligent Textile Equipment, Xi’an 710048, China; 3Xi’an Thermal Power Research Institute Co., Ltd., Xi’an 710054, China; 4State Key Laboratory of Metal Extrusion and Forging Equipment Technology, Xi’an 710032, China

**Keywords:** isothermal compression, B_4_C_p_/6063Al composites, flow stress, constitutive equation

## Abstract

Isothermal-compression tests of B_4_C_p_/6063Al composites containing 20 vol.% B_4_C were performed using a Gleeble-3500 device, at strain rates ranging from 0.001 s^−1^ to 1 s^−1^ and deformation temperatures ranging from 723 K to 823 K. The results showed that the high-temperature flow stress of B_4_C_p_/6063Al composites increases with the decrease in deformation temperature or the increase in the strain rate. After friction correction, the friction corrected stress was less than the original experimental stress. At the initial stage of deformation, the difference between the rheological stress after friction correction and the measured rheological stress is small, but with the continuous increase in the strain, the difference between the rheological stress after friction correction and the measured rheological stress is grows. Under the same strain, the difference between the rheological stress before and after friction correction becomes more significant with the decrease in the deformation temperature and the increase in the strain rate. Next, the material constants (i.e., α, β, Q, A, n) of B_4_C_p_/6063Al composites were calibrated based on the experimental data, and a constitutive equation was established based on Arrhenius theory. The experimental values and predicted values of the stress–strain curves are in good agreement with the stress–strain curves of the finite element simulation, and the validity of the constitutive equation was verified.

## 1. Introduction

Metal matrix composites are extensively used in engineering materials due to their superior mechanical properties over traditional metals. A variety of metal matrix composite materials, because of their low density and high strength, have attracted attention. The wear resistance, stiffness, hardness, and other properties of aluminum alloy are significantly enhanced by adding hard, reinforced particles [1,2]. Generally, the hard composites added in the aluminum alloy matrix include SiC, Al_2_O_3_, TiC, and B_4_C, etc. Among these, B_4_C has low density (2.51 g/cm^3^), high hardness and high strength, a small thermal expansion coefficient, and good chemical stability. In addition, B_4_C-particle-reinforced aluminum matrix composites have good neutron-absorption properties and can be used to make neutron-absorber plates.

The research on B_4_C_p_/Al composites mainly includes the preparation process [3], strengthening mechanism [4], and mechanical properties [1,5,6,7] of B_4_C_p_/Al composites. Liu et al. discussed the effect of B_4_C_p_ content on the high-temperature oxidation resistance of the composite. The results showed that the more B_4_C_p_ content in the material, the better the high-temperature oxidation resistance [8]. Guo et al. discussed the influencing factors on the interfacial bonding strength between B_4_C particles and Al during the preparation of B_4_C/Al and studied the strengthening and toughening mechanism of the composite [9]. However, there are few studies on the thermoplastic properties and flow field properties of B_4_C_p_/Al composites. The existence of B_4_C particles greatly reduces the ductility of the composite material; its thermoplasticity is quite different from that of the matrix; and its thermal-deformation properties are quite different from that of the matrix [6,7,8,9,10]. Studying the hot-deformation characteristics of boron-carbide-particle-reinforced aluminum matrix composites can help improve understanding of its rheology and plastic deformation, which has great practical significance for the processing of B_4_C-particle-reinforced aluminum matrix composites. Gao et al. studied the effect of hot pressing temperature on the microstructure and mechanical properties of B_4_C_p_/6061Al composites and obtained the effect of temperature on the hardness and conductivity of the material [11]. Wu et al. studied the effect of extrusion at different temperatures on the microstructure and mechanical properties of B_4_C-particle-reinforced aluminum composite. The results showed that hot extrusion had a significant positive effect on the improvement of the mechanical properties of the composite [12].

Generally, the deformation process can be described by the constitutive relationship of the material; that is, the relationship between temperature, strain rate, and strain in the deformation process is established from the experimental data. At present, formulas such as the Arrhenius, Zerrilli–Armstrong, and Rusinek–Klepaczko formulas are widely used to study the flow stress of materials [10]. Most commonly, the constitutive model of alloy materials including aluminum, titanium, magnesium, nickel, and niobium is based on the Arrhenius equation [13,14,15,16,17,18,19,20]. The model was proposed by Rokni and Zarei-Hanzaki to describe the flow stress of materials using the Arrhenius equation, which is suitable for a wide range of stresses.

Prior to this, S. Gangolu et al. [21] researched the flow characteristics of Al-5 wt.% B_4_C composites through compression tests performed within a specific temperature range (i.e., from 200 °C to 500 °C) and strain rate range (i.e., from 10^−4^ s^−1^ to 100 s^−1^). The optimum processing conditions for the Al-5 wt.% B_4_C composites were as follows: the strain rate was 10^−4^ s^−1^, the temperature range was from 425 °C to 475 °C, and the constitutive equation of Sellars-McG Tegart based on strain compensation was established [21]. Following this, compression experiments were carried out on Al-6.65Si-0.44Mg (A356) alloys and A356 + 5 wt.% B_4_C, and numerical simulations were carried out. Stability and instability changes were made to the process diagrams of A356 alloy and A356 + 5 wt.% B_4_C [22]. Liu et al. studied the high-temperature flow properties of 25 vol.% B_4_C_p_/2009 Al composites by isothermal-compression experiments, and a constitutive model based on the Arrhenius scheme was proposed and verified [6]. Zhou et al. considered the high strain rate correction to its constitutive equation, corrected the experimental flow stress by using the Arrhenius factor related to temperature, and verified the accuracy of the constitutive model by using the simulation results [23].

In this study, the flow stress of B_4_C_p_/6063 Al composites was investigated by uniaxial isothermal-compression experiments. The friction correction equation proposed by R.EbrahimiA and Najafizadeh corrects the flow stress and considers the effect of friction at different stages on the flow stress of the B_4_C_p_/6063 Al composite. Through the test results, the material constants of B_4_C_p_/6063 Al composites were obtained, the constitutive relation was deduced based on Arrhenius, and the parameters of the constitutive equation were calibrated. Using the established constitutive model equation to establish the finite element model to analyze the plastic deformation characteristics of a B_4_C_p_/6063 Al composite, the simulation results are in good agreement with the experimental results, which proves the accuracy of the constitutive model equation.

## 2. Experiments: A Theoretical Basis Is Provided for the Plastic Deformation Characteristics of B_4_C_p_/6063 Al Composites

### 2.1. Materials

The B_4_C_p_/6063Al composites containing 20 vol.% B_4_C particles were produced by the powder metallurgy method in the present investigation. The cylindrical specimen used in the isothermal-compression test was 10 mm in diameter and 15 mm in height, as shown in Figure 1, and graphite lubricant was used during the experiment to reduce the friction between the dies and the specimen end faces. The original morphology of the sample was observed using a metallographic microscope as shown in Figure 2. The B_4_C particles were evenly dispersed throughout the aluminum alloy matrix. The chemical composition of 6063 aluminum alloy in mass fraction is shown in Table 1.

### 2.2. Methods

The true stress–strain curves of the sample were discovered by a Gleeble-3500 gadget. The gadget is composed of a heating system, mechanical system, and digital control system. The maximum heating rate can reach 10,000 °C/s, the temperature control accuracy can reach ±1 °C, the maximum cooling rate can reach 10,000 °C/s on the sample surface, the maximum tensile and compressive static loads are 100 KN, the maximum axial displacement rate is 1000 mm/s, the minimum axial controllable rate is 0.01 mm/s, and the displacement measurement accuracy is 0.002 mm. Isothermal compression was tested at strain rates of 0.001 s^−1^, 0.01 s^−1^, 0.1 s^−1^, and 1 s^−1^, as well as temperatures of 723 K, 748 K, 773 K, 798 K, and 823 K. The sample’s height was reduced to 60%. During the experiment, the specified temperature was reached at a heating rate of 278 K/s. The deformation temperature of the sample was measured by a platinum–rhodium thermocouple that was welded to the center region of the sample surface. A heating rate that is too fast will cause deformation and warping of the sample, so the sample was heated to the required temperature and maintained there for 3 min before isothermal compression. At the end of the test, the sample was quenched in water. The experimental steps of the thermal compression test were shown in Figure 3. The axial surface of the sample was selected to observe the microstructure properties. The metallographic sample was prepared by chemical etching and mechanical polishing. The chemical etchant consisted of 95 vol.% H_2_O, 2.5 vol.% HNO_3_, 1.5 vol.% HCl, and 1 vol.% HF. The microstructure of the sample was observed by Leitz3DMIXT optical microscope.

## 3. Results and Discussion

### 3.1. Friction Correction

During the isothermal-compression tests, the lubricant was able to reduce the friction between the dies and specimens, but the friction became increasingly evident due to the area of interface increasing. The experimental results are influenced by the friction and the thermal effect of deformation, which will cause an increase in the size of the stress error, meaning that the flow stress curve cannot accurately reflect the plastic deformation of the material. Especially for the constitutive equation calibration, the accuracy of experimental results is a prerequisite. Therefore, a friction correction of the flow stress was performed to reduce the experimental stress error. The friction correction equation was proposed by Ebrahimi and Najafizadeh [24], and the measured flow stress was corrected by the following Equation (1). The shape of the sample before and after compression is shown in Figure 4.
(1)$$\sigma =\frac{P{\left(\frac{2mR}{H}\right)}^{2}}{2\left[\exp\left(\frac{2mR}{H}\right)-\frac{2mR}{H}-1\right]}$$

Here, *P* is the true stress before correction; *σ* is the corrected flow stress; *R* and *H* are the instantaneous values of the radius and height of samples, which can be calculated according to Equations (2) and (3), respectively; and m is the friction factor calculated according to Equation (4).
(2)$$R=R_{0}\exp\left(\varepsilon /2\right)$$
(3)$$H=H_{0}\exp\left(-\varepsilon \right)$$
(4)$$m=\frac{\left(R_{f}/H_{1}\right)b}{\left(4/\sqrt{3}\right)-\left(2b/3\sqrt{3}\right)}$$
(5)$$b=4\frac{\Delta R}{R_{f}}\cdot \frac{H_{1}}{\Delta H}$$

Here, *R*_0_ is the initial radius of samples, *ε* is the strain, *R_f_* is the average radius of specimens after compression, *b* is the barrel parameter, Δ*R* is the difference between the maximum radius and the top radius of deformed samples, and Δ*H* is the final height change of the samples after compression.
(6)$$R_{f}=R_{0}\sqrt{H_{0}/H_{1}}$$
(7)$$\Delta R=R_{M}-R_{T}$$
(8)$$R_{T}=\sqrt{3\frac{H_{0}}{H_{1}}R_{0}{}^{2}-2R_{M}^{2}}$$

Here, *R_M_* is the maximum radius of deformed samples, *R_T_* is the top radius of deformed samples, *H*_0_ is the initial height of samples, and *H*_1_ is the final height of samples.

### 3.2. Flow Stress Behavior

The solid lines in Figure 5 show the true stress–strain curves of B_4_C_p_/6063Al composites under different deformation temperatures. It can be seen from Figure 5 that the deformation behavior of B_4_C_p_/6063Al composites can be roughly divided into two stages. In the initial stage, the flow stress increases rapidly to the peak as the strain increases. The main reason is that the dislocation density increases sharply, and dislocation motion accelerates during the deformation process of the material. Additionally, dislocation motion limited to a certain range cannot easily overcome the obstacles, dislocation tangles, pinning, and B_4_C particles, which results in work hardening.

Subsequently, after the flow stress reaches the peak value, the flow stress decreases with the increase in strain. The primary explanation is that work hardening can be partially or completely compensated for by dynamic softening processes, such as dynamic recovery (DRV) or dynamic recrystallization (DRX), which results in a reduction in flow stress. As can be considered from Figure 5, the flow characteristics of B_4_C_p_/6063Al composites are sensitive to temperature and strain rate. Moreover, the flow stress gradually decreases with the increase in the deformation temperature, while the flow stress will increase swiftly with the increase in the stress rate at the identical deformation temperature. With the increase in deformation temperature, the kinetic energy of metal is increased, which makes the dynamic recrystallization or dynamic recovery fully occur, thereby enhancing the softening effect of the material. As a result, the flow stress is reduced. The work-hardening rate increases with the increase in the strain rate at the same deformation temperature. At the same time, the increase in the strain rate shortens the time of DRV or DRX, so that the softening cannot be fully carried out. As a result, the flow stress is increased.

### 3.3. Constitutive Equation of B_4_C_p_/6063Al Composite

The constitutive equation of B_4_C_p_/6063Al composite was developed to describe the deformation behavior of samples under different temperatures and different strain rates as well as the effects of deformation conditions on flow stress. Under different stress levels, the flow stress and strain rate of materials conform to the following relations [25,26].
(9)$$\dot{\varepsilon }=A_{1}\sigma^{n_{1}}\;\;\;\;\;(\mathrm{for}\;\mathrm{low}\;\mathrm{stress}\;\mathrm{level})$$
(10)$$\dot{\varepsilon }=A_{2}\exp\left(\beta \sigma \right)\;\;(\mathrm{for}\;\mathrm{high}\;\mathrm{stress}\;\mathrm{level})$$

Here, *A*_1_, *A*_2_, and *β* are material constants, *n*_1_ is the stress exponent, $\dot{\varepsilon }$ is the strain rate (s^−1^), and *σ* is flow stress (MPa). The two formulas can be unified into the hyperbolic sine formula, as follows: (11)$$\dot{\varepsilon }=A{\left[\sinh\left(\alpha \sigma \right)\right]}^{n}\exp\left(-Q/RT\right)\;\;\;\left(\mathrm{for}\;\mathrm{all}\;\sigma \right)$$

Here, *α* is the material constant, *Q* is the effective activation energy for deformation (J. Mol^−1^), *R* is the universal gas constant, 8.314 J. Mol^−1^ K^−1^, and *T* is the absolute temperature (K). *β* and *α* exist in the relationship of *α* = *β*/*n*_1_.

The friction-corrected true stress–strain curves were used to calculate the material parameters of the B_4_C_p_/6063Al composite constitutive equation. The natural logarithm of Equations (9) and (10) were taken, respectively, as shown in Equations (12) and (13):(12)$$\ln\dot{\varepsilon }=\ln A_{1}+n\ln\sigma$$
(13)$$\ln\dot{\varepsilon }=\ln A_{2}+\beta \sigma$$

According to the changes in the peak flow stress, the figures of $\ln\dot{\varepsilon }-\ln\sigma$ and $\ln\dot{\varepsilon }-\sigma$ can be plotted for different temperatures. As shown in Figure 6, the natural logarithm of the strain rate shows a significant linear correlation with the natural logarithm of the peak stress; similarly, the natural logarithm of the strain rate also has a significant linear correlation with the peak stress. According to Equations (12) and (13), the slopes of the lines in Figure 6 gives an approximate value for *n* and *β*. The linear regression effects show that the stress exponent *n* and the material consistent *β* are about 11.397 and 0.123. This gives a material constant *α* of about 0.011.

Equation (14) can be obtained through the logarithm of Equation (11).
(14)$$\ln\dot{\varepsilon }=\ln A+n\ln\sinh\left(\alpha \sigma \right)-Q/RT$$

When the material constant (*A*, *Q*, α, *n*) is obtained, the variation law of the flow stress of the alloy under high-temperature plastic deformation can be obtained. When the stress level is unchanged, there is a linear relationship between $\ln{\dot{\varepsilon }}_{s}$ and 1/T. Here, ${\dot{\varepsilon }}_{s}$ is the strain ratio. According to the experimental results of deformation at different temperatures and strain rates, $n=n_{1},\alpha =\beta /n$ can be obtained. When aluminum and aluminum alloy composite materials are used, the value and applied stress are very small and can be ignored. Equation (15) can be obtained.
(15)$$\dot{\varepsilon }=A\sigma n\exp(Q/RT)$$

Partial differentiation of Equation (15) leads to Equation (16):(16)$$Q=R{\left[\frac{\partial \ln\dot{\varepsilon }}{\partial \ln\sigma }\right]}_{T}{\left[\frac{\partial \ln\sigma }{\partial 1/T}\right]}_{\dot{\varepsilon }}$$

Here, ${\left[\frac{\partial \ln\dot{\varepsilon }}{\partial \ln\sigma }\right]}_{T}$ = *n*, and *n* is the stress exponent. For composite materials, σ can be substituted by sinh(ασ) in Equation (15), and Equation (17) can be obtained.
(17)$$Q=R{\frac{\partial \ln\dot{\varepsilon }}{\partial \ln\left[\sinh\left(\alpha \sigma \right)\right]}|}_{T}{\frac{\partial \left[\ln\sinh\left(\alpha \sigma \right)\right]}{\partial \left(1/T\right)}|}_{\dot{\varepsilon }}$$

It can be seen from Equation (17) that the values of $\partial \ln\dot{\varepsilon /}\partial \ln[\sin h\left(\alpha \sigma \right)]$ and ∂[lnsinh(ασ)]/∂(1/T) can be evaluated by plotting $\ln\dot{\varepsilon }-\ln[\sin h\left(\alpha \sigma \right)$ and lnsin(ασ)−1/T [27]. As shown in Figure 7, the graphs of $\ln\dot{\varepsilon }-\ln[\sin h\left(\alpha \sigma \right)$ and lnsin(ασ)−1/T have a significant linear correlation. The slopes of the lines in Figure 7 indicate the values for $\partial \ln\dot{\varepsilon /}\partial \ln[\sin h\left(\alpha \sigma \right)]$ and lnsin(ασ)−1/T, following Equation (17). There are roughly 8.483 and 442.5865 slopes on average, respectively. Equation (17) was modified to include the value of *R* and the two slope values, which resulted in a calculation showing that the average Q-value of B_4_C_p_/6063Al composites is approximately 312.146 KJ/mol.

When considering hot compression of composites, the Zener–Hollomon parameter can be expressed as Equation (18).
(18)$$Z=\dot{\varepsilon }\exp\left(Q/RT\right)=A{\left[\sinh\left(\alpha \sigma \right)\right]}^{n}$$

The natural logarithm of Equation (18) is given as:(19)$$\ln Z=\ln\dot{\varepsilon }+Q/RT=\ln A+n\ln\left[\sinh\left(\alpha \sigma \right)\right]$$

The value for *α* and *n* were introduced into Equation (18) to calculate the values of Z and  lnZ, and to plot lnZ−lnsinh(ασ). As shown in Figure 8, it can be seen that lnZ has a highly linear correlation with lnsinh(ασ). According to Equation (19), the exact *n*-values are derived from the slope of lnZ−lnsinh(ασ), and lnA is obtained from the intercept of lnZ−lnsinh(ασ). Figure 8 shows the plot of lnZ−lnsinh(ασ) with slope and intercept at about 8.394 and 43.333, respectively. Therefore, the values of the material constant A and the stress exponent *n* are 6.596 × 10^18^ and 8.394, respectively.

Finally, introducing the calculated material parameters into Equation (11), the constitutive equation of the B_4_C_p_/6063Al composite is extracted as follows:(20)$$\dot{\varepsilon }=6.596\times 10^{18}{\left[\sinh\left(0.011\sigma \right)\right]}^{8.394}\exp\left(-\frac{312.146}{RT}\right)$$

### 3.4. Verification of the Constitutive Equation

The rationality of constitutive equations directly affects the accuracy of the prediction of the rheological properties of materials. In this study, the validity of constitutive equations for B_4_C_p_/6063Al composites was verified from the comparison of flow stress peaks and finite element simulation [28,29,30].


**(1) Comparison of flow stress peaks of B_4_C_p_/6063Al composites**


Equation (21) is obtained by the transformation of Equation (18). Using a hyperbolic sine function conversion for Equations (21) and (22), flow stress on Z parameters can be obtained. Substituting the material parameters of the B_4_C_p_/6063Al composite into Equation (22) yields Equation (23), then the flow stress in different states can be calculated for corresponding deformation conditions. The predicted values of peak flow stress are obtained by substituting the strain rate and strain temperature into Equation (23) and the predicted and friction corrected values from the experiment are compared. As shown in Figure 9, there is little difference between the predicted value and the corrected value, and the maximum error of peak stress is 5.6%. Therefore, it can be proved that the constitutive equation of the B_4_C_p_/6063Al composites is valid.
(21)$$\sinh\left(\alpha \sigma \right)={\left(Z/A\right)}^{1/n}$$
(22)$$\sigma =\frac{1}{\alpha }\ln\left\{{\left(Z/A\right)}^{1/n}+{\left[{\left(Z/A\right)}^{2/n}+1\right]}^{1/2}\right\}$$
(23)$$\left\{ \begin{array}{l} \sigma =\frac{1}{0.011}\ln\left\{{\left(Z/6.596\times 10^{18}\right)}^{1/8.394}+{\left[{\left(Z/6.596\times 10^{18}\right)}^{2/8.394}+1\right]}^{1/2}\right\}\\ Z=\dot{\varepsilon }\exp\left[312.146/\left(8.314\times T\right)\right] \end{array}\right.$$


**(2) Finite element simulation of Gleeble upsetting tests**


The isothermal-compression process of B_4_C_p_/6063Al composites was simulated. The conditions of the numerical simulation are similar to those of the isothermal-compression test. The extrusion speed is determined by the strain rate. According to Equation (24), the flow stress of material during the simulation can be calculated [31].
(24)$$\left\{ \begin{array}{l} \sigma =\frac{F\left(H_{0}-\int_{0}^{t}v_{0}\exp\left(\dot{\varepsilon }\cdot t\right)dt\right)}{H_{0}\cdot S_{0}}\\ \varepsilon =\ln H_{n}/H_{0} \end{array}\right.$$

Here, *F* is the instantaneous axial load of the die in the simulation process; $H_{0}$ is the initial height of the blank; $v_{0}$ is the initial velocity of the top die; $v_{0}$ = $H_{0}\cdot \dot{\varepsilon }$, $S_{0}$ is the initial stress area of the blank; $\dot{\varepsilon }$ is the strain rate; *t* is the runtime of the top die; $\dot{\varepsilon }$ is the true strain; and *H_n_* is the instantaneous height of the blank.

The constitutive equation of B_4_C_p_/6063Al composite was introduced into the material library of numerical simulation software, and the compression test was simulated in the temperature range from 723 K to 823 K at the strain rate of 1 s^−1^.

Figure 10 illustrates the agreement between simulated and experimental data, indicating the validity of the constitutive equation for the B_4_C_p_/6063Al composite. The higher the temperature and strain, the higher the agreement between the simulation results and the experimental results, which is almost consistent with the true stress–strain law obtained in the literature at a specific temperature. At a certain deformation temperature, when the strain rate is high, the time required for the specimen to reach a certain amount of deformation is shortened, the dislocation proliferation rate is increased during the deformation process, and the dislocations are interlaced and entangled with each other, resulting in an increase in the critical stress required for deformation, and then higher flow stress. With the increase in strain, various softening mechanisms gradually weaken the hardening effect, and an obvious rheological softening phenomenon appears. When the strain rate is constant, with the increase in deformation temperature, the flow stress decreases and the degree of rheological softening weakens. As shown in Figure 10, since the strain is loaded from 0 in the simulation process, the actual stress yield point in the simulation results appears earlier than the experimental results, but the overall trend is consistent, which proves that the constitutive equation can be used for finite element simulations and is helpful to further study of the plastic forming properties of materials.

## 4. Conclusions

(1)For the thermal deformation of 20 vol.% B_4_C_p_6061al composite at a range of temperatures and a strain rate of 0.6 s^−1^, the flow stress decreases with the increase in temperature or strain rate, and it is proposed that this can be expressed by the Arrhenius constitutive equation.(2)As the influence of friction on flow stress becomes increasingly obvious with the increase in compression during the experiment, the influence of friction on flow stress should be corrected according to the experimental results. The Arrhenius constitutive constant of the 20 vol.% B_4_C_p_/6061Al composite was obtained according to the experimental data and the corrected data.(3)The reliability of the constitutive equation is verified by comparing the experimental values and predicted values of the peak flow stress of the B_4_C_p_/6063Al composite and the finite element simulation. Moreover, the constitutive equation can be used for finite element simulation, which is helpful for studying the plastic-forming performance of materials further.

## Figures and Tables

**Figure 1 materials-15-06438-f001:**
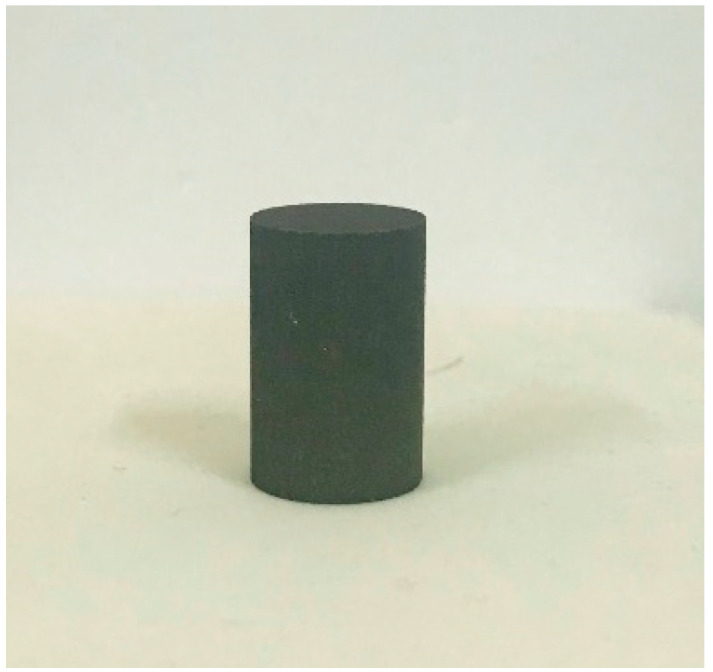
Testing Sample.

**Figure 2 materials-15-06438-f002:**
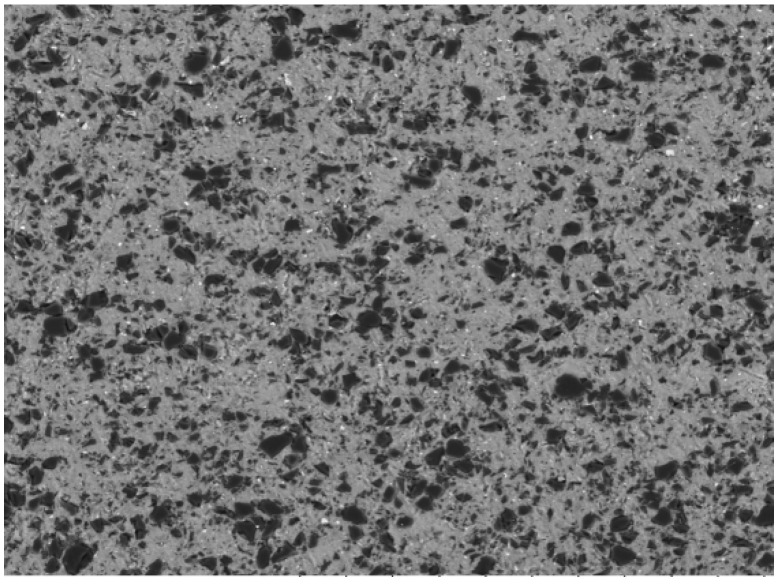
Initial microstructures of B_4_C_p_/6063Al composite.

**Figure 3 materials-15-06438-f003:**
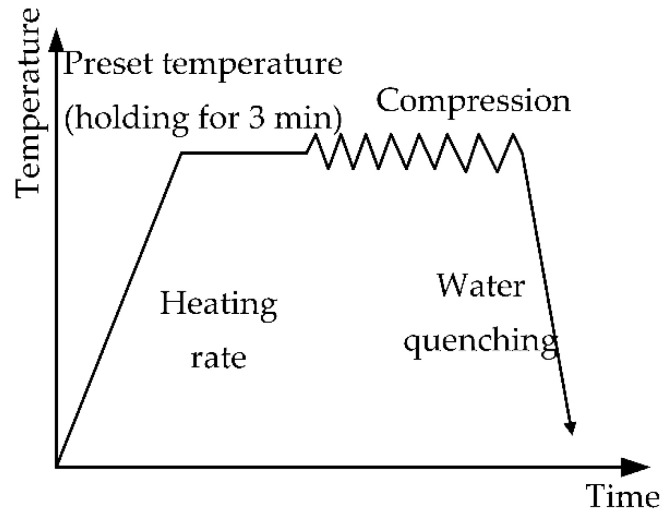
Solution process curve for hot-compression tests.

**Figure 4 materials-15-06438-f004:**
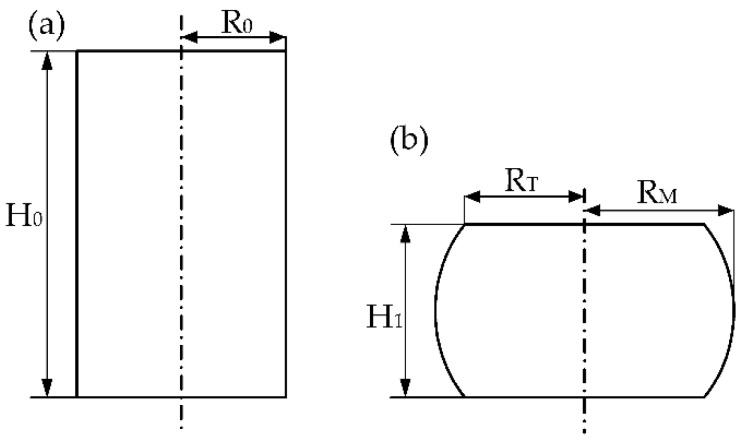
Diagrams of the sample before and after compression: (**a**) before compression; (**b**) after compression.

**Figure 5 materials-15-06438-f005:**
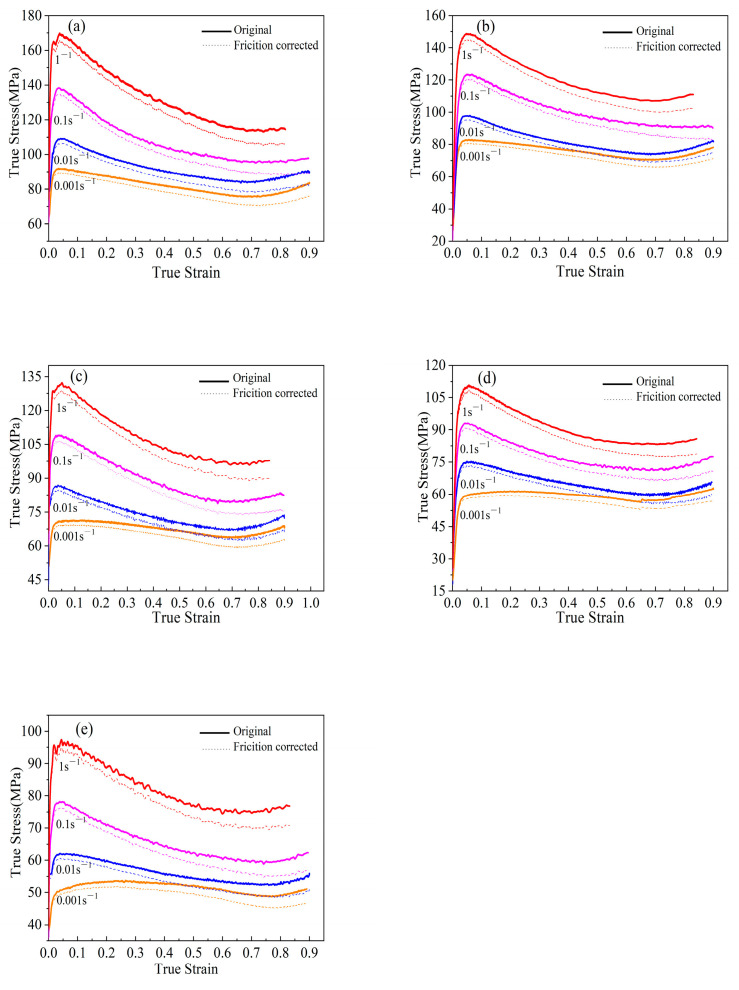
True stress–strain curves of B_4_C_p_/6063Al composite at the temperature of (**a**) 573 K, (**b**) 623 K, (**c**) 673 K, (**d**) 723 K, and (**e**) 773 K.

**Figure 6 materials-15-06438-f006:**
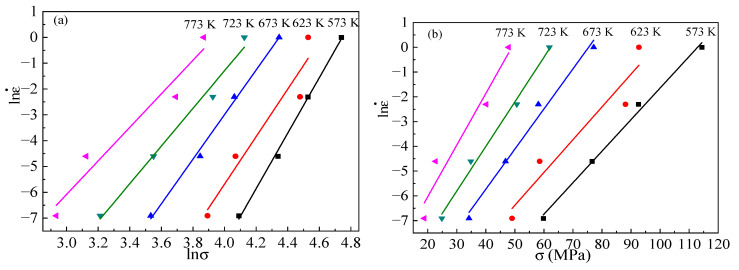
Relationships of (**a**) $\ln\dot{\varepsilon }-\ln\sigma$; (**b**) $\ln\dot{\varepsilon }-\sigma$.

**Figure 7 materials-15-06438-f007:**
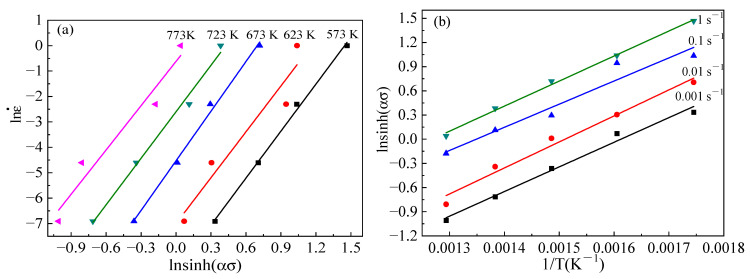
Relationship of (**a**) $\partial \ln\dot{\varepsilon }/\partial \ln\left[\sinh\left(\alpha \sigma \right)\right]$ and (**b**) ∂[lnsinh(ασ)]/∂(1/T) for B_4_C_p_/6063Al.

**Figure 8 materials-15-06438-f008:**
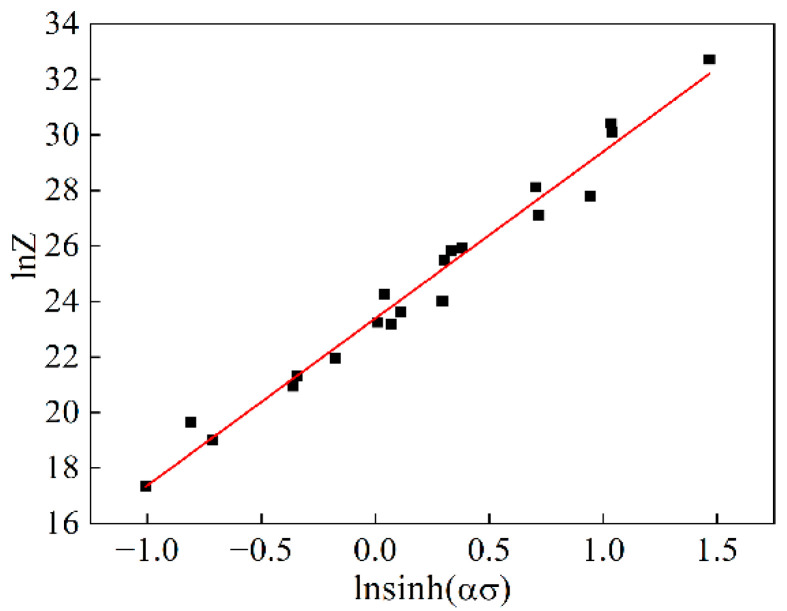
Relationships between Z parameter and flow stress.

**Figure 9 materials-15-06438-f009:**
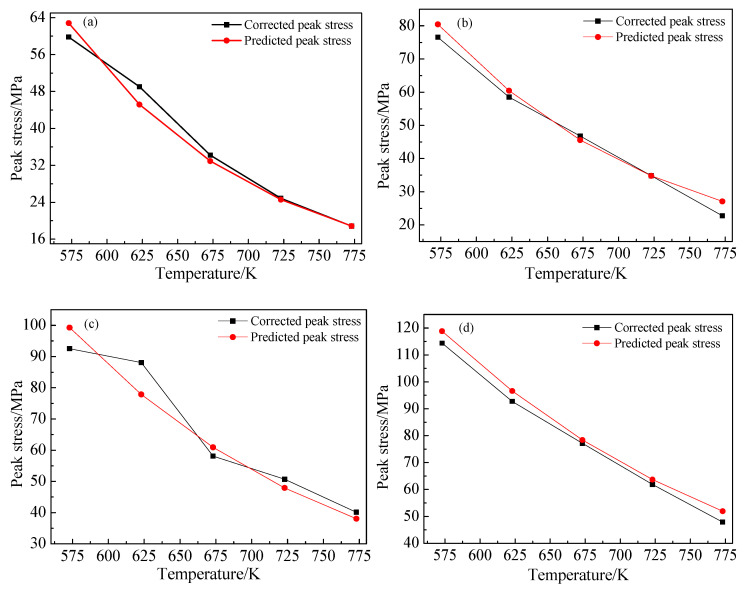
Comparison between the corrected and the predicted peak stress of B_4_C_p_/6063Al composites: (**a**) $\dot{\varepsilon }=0.001\;s^{-1}$, (**b**) $\dot{\varepsilon }=0.01\;s^{-1}$, (**c**) $\dot{\varepsilon }=0.1\;s^{-1}$, (**d**) $\dot{\varepsilon }=1\;s^{-1}$.

**Figure 10 materials-15-06438-f010:**
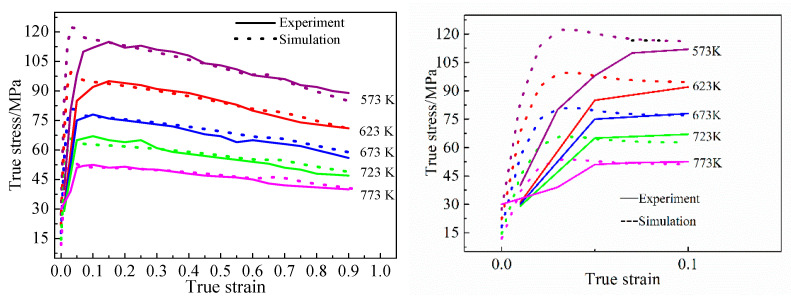
Comparison of simulated with experimental stress–strain curves and local amplification with strain less than 0.1.

**Table 1 materials-15-06438-t001:** Chemical compositions of 6063 aluminum alloy (mass fraction, %).

Si	Mg	Fe	Cu	Mn	Zr	Cr	Al
0.2–0.6	0.45–0.9	<0.35	<0.1	<0.1	<0.1	<0.1	Bal.

## Data Availability

Data sharing not applicable.

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
