# Peer review of "Calibration of Arrhenius Constitutive Equation for B4Cp/6063Al Composites in High Temperatures"

_materials, 2022, doi:10.3390/ma15186438_

Round 1

Reviewer 1 Report

In this manuscript authors made an attempt to compare the experimental (isothermal compression tests) and predicted values (using finite element simulation) of stress-strain curves at several temperatures and strain rates for the B4Cp/6063Al composites. The manuscript is well written and  discusses most of the aspects with clarity.

I suggest authors to improve the introduction by incorporating more relevant researches in the recent past as well (for example: Journal of Materials Science & Technology 34 (2018) 1730–1738 and more).

However, in Figure 9, I request authors to magnify the True stress behavior at lower True strain (below 0.1). It is very certain that at the higher temperature the data fitting for attaining the yield point is either possessing huge ambiguity or rather underestimates the same. Kindly include an explanation for the same and/or improve the curve fitting by incorporating the strain softening parameter.

A minor comment: please unify the formatting of the references. Currently, it is difficult to track down all of the references. 

Author Response

Response to Reviewer 1 Comments

Dear Editors and Reviewer #1:

Thank you for your letter and for the reviewers’ comments concerning our manuscript entitled “Calibration of Arrhenius Constitutive Equation for B4Cp/6063Al Composites in High Temperatures” (ID: 1858685). Those comments are all valuable and very helpful for revising and improving our paper, as well as the important guiding significance to our researches. We have studied comments carefully and have made correction which we hope meet with approval. Revised portion are marked in red in the paper. The main corrections in the paper and the responds to the reviewer’s comments are as flowing:

Responds to the reviewer’s comments:

Point 1: I suggest authors to improve the introduction by incorporating more relevant researches in the recent past as well (for example: Journal of Materials Science & Technology 34 (2018) 1730–1738 and more).

Response 1: As Reviewer suggested that in the introduction part of the article, we made a lot of amendments to the introduction, added references in recent years, including articles recommended by reviewers, and classified the literature to make the introduction more clear and forward-looking, and gave a more detailed description of this study. As shown in line 42-49, 56-62, 63-76, 82-96 marked in red.

Point 2: However, in Figure 9, I request authors to magnify the True stress behavior at lower True strain (below 0.1). It is very certain that at the higher temperature the data fitting for attaining the yield point is either possessing huge ambiguity or rather underestimates the same. Kindly include an explanation for the same and/or improve the curve fitting by incorporating the strain softening parameter.

Response 2: We have made correction according to the Reviewer’s comments. We enlarged and explained Figure 10 (original figure 9). As shown in line 287-305 marked in red.

Point 3: A minor comment: please unify the formatting of the references. Currently, it is difficult to track down all of the references.

Response 3: We supplemented and reordered the references.

In addition, other modifications to the article are as follows:

(1) We supplemented and reconstructed the abstract of the article and added some conclusions to make the abstract more compact and reasonable. As shown in line 18-23, 25-27 marked in red.

(2) We have explained the innovation of the article in more detail. The innovation of this paper is to study the flow stress of B4Cp / 6063 aluminum composite through uniaxial isothermal compression experiment. R. The friction correction equation proposed by ebrahimia and najafizadeh corrects the flow stress and considers the effect of friction at different stages on the flow stress of B4Cp / 6063 aluminum composite. Through the test results, the material constants of B4Cp / 6063 aluminum composite were obtained, and the constitutive relationship was derived based on Arrhenius, and the parameters of the constitutive equation were calibrated. The finite element model is established by using the established constitutive model equation to analyze the plastic deformation characteristics of B4Cp / 6063 aluminum composite. The simulation results are in good agreement with the experimental results, which proves the accuracy of the constitutive model. As shown in line 88-96 marked in red.

(3) we have added more details to the experimental part, such as the physical size photos of the prepared samples and the introduction of the experimental equipment. As shown in line 101-104, 108-109, 114-120, 122-133 marked in red.

(4) We added the comparison between the experimental and simulation results of this paper and the results of literature [23], which shows that the constitutive model proposed in this paper can well simulate the plastic forming performance of B4Cp / 6063 aluminum composite. As shown in line 291-297 marked in red.

(5) we reconstructed the conclusion part of the article and summarized the relevant conclusions to make the conclusion part of the article more specific and consistent with the research content of the article. As shown in line 299-312 marked in red.

(6) We reviewed and revised the grammar and language expression of the full text to make the article more fluent.

Special thanks to you for your good comments.

We appreciate for Editors/Reviewers’ warm work earnestly, and hope that the correction will meet with approval.

Once again, thank you very much for your comments and suggestions.

Reviewer 2 Report

Manuscript Number: materials-1858685
Manuscript title: Calibration of Arrhenius Constitutive Equation for B4Cp/6063Al Composites in High Temperatures

The authors should put a tremendous effort into technically and linguistically shaping the contents of the manuscript. My comments are listed as follows:
- The whole abstract was not properly structured, and more results should be added.
-The introduction could be much improved and recent researches should be added.
-The novelty of the work is missing in the manuscript till now.
-The experimental program must be reconstructed. It is advised to include some photos of sample preparation for a better understanding of the test preparation and technical drawings of the test set up including all the instrumentation and samples dimensions.
- There are several studies reporting similar data but the authors offer no comparison with these results (I didn't find any comparison of results with past studies).

- Meaningful conclusions are needed as conclusions are general and are common sense. The conclusion must be reconstructed & simplified into 4-5 main important points.

-There are still some grammar problems, which need to be carefully checked throughout the whole article

Author Response

Response to Reviewer 2 Comments

Dear Editors and Reviewer #2:

Thank you for your letter and for the reviewers’ comments concerning our manuscript entitled “Calibration of Arrhenius Constitutive Equation for B4Cp/6063Al Composites in High Temperatures” (ID: 1858685). Those comments are all valuable and very helpful for revising and improving our paper, as well as the important guiding significance to our researches. We have studied comments carefully and have made correction which we hope meet with approval. Revised portion are marked in red in the paper. The main corrections in the paper and the responds to the reviewer’s comments are as flowing:

Responds to the reviewer’s comments:

Point 1: The whole abstract was not properly structured, and more results should be added.

Response 1: Considering the Reviewer’s suggestion, we supplemented and reconstructed the abstract of the article and added some conclusions to make the abstract more compact and reasonable. As shown in line 18-23, 25-27 marked in red.

Point 2: The introduction could be much improved and recent researches should be added.

Response 2: We have made correction according to the Reviewer’s comments. we made a lot of amendments to the introduction, added references in recent years, including articles recommended by reviewers, and classified the literature to make the introduction more clear and forward-looking, and gave a more detailed description of this study. As shown in line 42-49, 56-62, 63-76, 82-86 marked in red.

Point 3: The novelty of the work is missing in the manuscript till now.

Response 3: The innovation of this paper is to study the flow stress of B4Cp / 6063 aluminum composite through uniaxial isothermal compression experiment. R. The friction correction equation proposed by R.EbrahimiA and Najafizadeh corrects the flow stress and considers the effect of friction at different stages on the flow stress of B4Cp / 6063 aluminum composite. Through the test results, the material constants of B4Cp / 6063 aluminum composite were obtained, and the constitutive relationship was derived based on Arrhenius, and the parameters of the constitutive equation were calibrated. The finite element model is established by using the established constitutive model equation to analyze the plastic deformation characteristics of B4Cp / 6063 aluminum composite. The simulation results are in good agreement with the experimental results, which proves the accuracy of the constitutive model. As shown in line 88-96 marked in red.

Point 4: The experimental program must be reconstructed. It is advised to include some photos of sample preparation for a better understanding of the test preparation and technical drawings of the test set up including all the instrumentation and samples dimensions.

Response 4: Considering the Reviewer’s suggestion, we have added more details to the experimental part, such as the physical size photos of the prepared samples and the introduction of the experimental equipment. As shown in line 101-104, 108-109, 114-120, 122-133 marked in red.

Point 5: There are several studies reporting similar data but the authors offer no comparison with these results (I didn't find any comparison of results with past studies).

Response 5: We have rewritten this part according to the Reviewer’s suggestion. We added the comparison between the experimental and simulation results of this paper and the results of literature [23], which shows that the constitutive model proposed in this paper can well simulate the plastic forming performance of B4Cp / 6063 aluminum composite. As shown in line 291-297 marked in red.

Point 6: Meaningful conclusions are needed as conclusions are general and are common sense. The conclusion must be reconstructed & simplified into 4-5 main important points.

Response 6: As Reviewer suggested that in the conclusion part of the article, we reconstructed the conclusion part of the article and summarized the relevant conclusions to make the conclusion part of the article more specific and consistent with the research content of the article. As shown in line 299-312 marked in red.

Point 7: There are still some grammar problems, which need to be carefully checked throughout the whole article.

Response 7: We reviewed and revised the grammar and language expression of the full text to make the article more fluent.

In addition, other modifications to the article are as follows:

(1) We supplemented and reordered the references.

(2) We enlarged and explained Figure 10 (original figure 9). As shown in line 287-305 marked in red.

Special thanks to you for your good comments.

We appreciate for Editors/Reviewers’ warm work earnestly, and hope that the correction will meet with approval.

Once again, thank you very much for your comments and suggestions.

Round 2

Reviewer 1 Report

Authors have positively considered the suggestions and made changes in the manuscript for improving the manuscript structure. I suggest following minor suggestions:

1. Unify the references styling in recommended format by MDPI.

2. In the experimental section, please write chemicals names with correct subscripts for used acids , water etc. For instance, in HNO3, the "3" should be in subscript.

Author Response

Response to Reviewer 1 Comments

Dear Editors and Reviewer #1:

Thank you for your letter and for the reviewers’ comments concerning our manuscript entitled “Calibration of Arrhenius Constitutive Equation for B4Cp/6063Al Composites in High Temperatures” (ID: 1858685). Those comments are all valuable and very helpful for revising and improving our paper, as well as the important guiding significance to our researches. We have studied comments carefully and have made correction which we hope meet with approval. Revised portion are marked in red in the paper. The main corrections in the paper and the responds to the reviewer’s comments are as flowing:

Responds to the reviewer’s comments:

Point 1: Unify the references styling in recommended format by MDPI.

Response 1: We modified the reference format of the article according to the requirements of MDPI to make it meet the relevant requirements. As shown in line 131-132 marked in red.

Point 2: In the experimental section, please write chemicals names with correct subscripts for used acids , water etc. For instance, in HNO3, the "3" should be in subscript.

Response 2: We have made correction according to the Reviewer’s comments. We revised the names of the chemicals in the experimental part of the article and checked the full text. As shown in line 332-398 marked in red.

Special thanks to you for your good comments.

We appreciate for Editors/Reviewers’ warm work earnestly, and hope that the correction will meet with approval.

Once again, thank you very much for your comments and suggestions.

Reviewer 2 Report

Good revised work. The authors almost covered all the points, and the paper is accepted for publication.

Good luck

Author Response

Response to Reviewer 2 Comments

Dear Editors and Reviewer #2:

First of all, thank you for your letter and for the reviewers’ comments concerning our manuscript entitled “Calibration of Arrhenius Constitutive Equation for B4Cp/6063Al Composites in High Temperatures” (ID: 1858685). In the process of improving the paper, we also made some modifications to the article are as follows:

  1. We modified the reference format of the article according to the requirements of MDPI to make it meet the relevant requirements. As shown in line 131-132 marked in red.

2. We revised the names of the chemicals in the experimental part of the article and checked the full text. As shown in line 332-398 marked in red.
